# Elastic Band Training Versus Multicomponent Training and Group-Based Dance on Morphological Variables and Physical Performance in Older Women: A Randomized Controlled Trial

**DOI:** 10.3390/life14111362

**Published:** 2024-10-24

**Authors:** Jordan Hernandez-Martinez, Eduardo Guzmán-Muñoz, Izham Cid-Calfucura, Francisca Villalobos-Fuentes, Daissy Diaz-Saldaña, Ignacia Alvarez-Martinez, María Castillo-Cerda, Tomás Herrera-Valenzuela, Braulio Henrique Magnani Branco, Pablo Valdés-Badilla

**Affiliations:** 1Department of Physical Activity Sciences, Universidad de Los Lagos, Osorno 5290000, Chile; jordan.hernandez@ulagos.cl (J.H.-M.); franciscabelen.villalobos@alumnos.ulagos.cl (F.V.-F.); daissyyaqueline.diaz@alumnos.ulagos.cl (D.D.-S.); ignaciacatalina.alvarez@alumnos.ulagos.cl (I.A.-M.); acastill@ulagos.cl (M.C.-C.); 2Programa de Investigación en Deporte, Sociedad y Buen Vivir, Universidad de los Lagos, Osorno 5290000, Chile; 3School of Kinesiology, Faculty of Health, Universidad Santo Tomás, Talca 3530000, Chile; eguzmanm@santotomas.cl; 4School of Kinesiology, Faculty of Health Sciences, Universidad Autónoma de Chile, Talca 3530000, Chile; 5Department of Physical Activity, Sports and Health Sciences, Faculty of Medical Sciences, Universidad de Santiago de Chile (USACH), Santiago 8370003, Chile; izham.cid@gmail.com (I.C.-C.); tomas.herrera@usach.cl (T.H.-V.); 6Graduate Program in Health Promotion, Cesumar University (UniCesumar), Maringá 87050-900, Brazil; braulio.branco@unicesumar.edu.br; 7Department of Physical Activity Sciences, Faculty of Education Sciences, Universidad Católica del Maule, Talca 3530000, Chile; 8Sports Coach Career, School of Education, Universidad Viña del Mar, Viña del Mar 2520000, Chile

**Keywords:** exercise, resistance training, dancing, older adults, aging

## Abstract

Background: This study aimed to analyze the effects of elastic band training (EBT) versus multicomponent training (MCT) and group-based dance (GBD) on waist circumference, body composition (body fat percentage and fat-free mass), and physical performance (handgrip strength, HGS; 30-s chair stand; timed up-and-go, TUG) in Chilean older women. Methods: This is a randomized controlled trial with three parallel groups: EBT (*n =* 10), MCT (*n =* 10), and GBD (*n =* 10). Two 60-min sessions per week for 8 weeks were dedicated to the interventions with pre- and post-assessments. A two-factor mixed ANOVA model with repeated measures was performed to measure the time × group effect. Results: Multiple comparisons revealed significant differences between EBT and MCT in the body fat percentage (*p* = 0.001; ES = 2.488, *large effect*) in favor of MCT, while HGS in the non-dominant hand (*p* = 0.044; ES = 0.158) was in favor of EBT. In the intragroup results, only the MCT significantly decreased the body fat percentage (*p* = 0.044; ES = 0.426, *small effect*), and EBT significantly increased HGS in the dominant (*p* < 0.001; ES = 0.977, *large effect*) and non-dominant (*p* < 0.001; ES = 0.583, *moderate effect*) hands and improved the 30-s chair stand (*p* = 0.003; ES = 1.612, *large effect*) test. The GBD did not report significant changes. Conclusions: MCT significantly reduced the body fat percentage regarding EBT, and EBT significantly improved HGS in the non-dominant hand regarding MCT, with no differences reported in the rest of the analyzed variables between the groups.

## 1. Introduction

During aging, changes occur in body composition [1] and physical performance [2] characterized by an increase in total body fat mass, along with a decrease in fat-free mass and bone mineral density [1], walking speed and balance [2], muscle strength, and power [3]. Given this, physical inactivity potentiates these alterations in body composition and physical performance [4], increasing the risk of functional dependence and affecting the health-related quality of life in older people [5].

On the contrary, physical activity has positively affected body composition and physical performance in older people [4]. In a study by Hernandez-Martinez, et al. [4], it was observed that physically active older women had a significantly lower body fat percentage (*p* = 0.000), higher fat-free mass (*p* < 0.001), greater handgrip strength (HGS) (*p* < 0.001), and better performance in timed up-and-go (TUG) (*p* = 0.000) and walking speed (*p* < 0.01) compared to physically inactive older women. It is, therefore, essential to carry out interventions that increase physical activity levels in older people [6]. A rapid review conducted by Pinheiro, et al. [6] analyzed the impact of physical activity programs and services for older people worldwide, showing a 77% increase in physical performance through structured physical exercise compared to recreational physical activity, which improves physical performance by 36%. In structured exercise, multicomponent training (MCT) leads to an 81% improvement in physical performance, a 20% improvement in resistance training, and a 7% improvement in recreational physical activity, such as group-based dance (GBD), in older people [6].

The most traditionally used interventions in older people are MCT, which includes at least three essential physical qualities or skills, typically muscle strength, endurance, balance and/or flexibility, and GBD interventions to improve health status and physical performance [6,7]. There are novel, easily accessible, and inexpensive alternatives to traditional resistance training interventions, such as elastic band training (EBT) [8]. In a meta-analysis by Hernandez-Martinez, et al. [9] in older people, significant improvements in 30-s chair stand (*p* = 0.04), sit-and-reach (*p* = 0.04), and TUG (*p* < 0.0001) were reported by EBT interventions compared to active/inactive control groups. In a study carried out by Gargallo, et al. [10] in older women, they compared the effects of MCT versus EBT with 20-week interventions with a frequency of 2 sessions per week of 70 to 80 min duration per session. A significant decrease in body fat percentage (*p* < 0.001) and waist circumference (WC) (*p* < 0.001), along with significant improvements in physical performance of TUG (*p* < 0.001), HGS (*p* = 0.006), and the 30-s chair stand test (*p* < 0.001) were observed compared to an inactive control group. Another study conducted by Miranda-Aguilar, et al. [11] in older people compared EBT versus resistance training with traditional gym equipment for 6 weeks with a frequency of 2 sessions a week of 60 min duration, showing significant improvements in favor of traditional gym equipment in the HGS of the dominant hand (*p* = 0.001) and TUG (*p* = 0.04) in favor of EBT. Similar to that reported by Valdés-Badilla, et al. [12], in older women with sarcopenia, when comparing EBT versus GBD in 12-week interventions with a frequency of 3 sessions of 60 min per week, significant improvements were reported in favor of EBT in fat-free mass (*p* < 0.001), HGS in the dominant (*p* = 0.006) and non-dominant (*p* = 0.009) hands, and TUG (*p* = 0.008).

While there is evidence of the effects of MCT, EBT, and GBD on body composition and physical performance in older people [10,11,12], the beneficial effects of these interventions using a three-arm randomized controlled trial are unknown. Considering that older women present a higher risk of functional dependence than males in Chile [13] and three-arm randomized controlled trials help to provide greater transparency and accuracy in the presentation of outcomes to generate future recommendations [14], this study analyzed the effects of EBT versus MCT and GBD on WC, body composition (body fat percentage and fat-free mass), and physical performance (HGS, 30-s chair stand, and TUG) in Chilean older women. It is hypothesized that EBT and MCT lead to significantly improved physical performance and body composition compared to GBD [10,11,12].

## 2. Methods

### 2.1. Study Design

The study design included a three-arm randomized controlled trial (EBT group, MCT group, and GBD) where a card-performed randomization draw was performed with a single-blinded design (evaluators blinded). The randomization was made using the randomizer internet site (https://www.randomizer.org). The methodology followed was the CONSORT guidelines [15]. In addition, it has been registered in the Clinical Trial Protocol Registry and Results System of the United States of America (code: NCT05275140; https://clinicaltrials.gov/search?cond=NCT05275140, first posted on 11 March 2022). The interventions were carried out over 8 weeks and comprised 16 sessions. These sessions were conducted twice a week (Tuesday and Thursday), lasting 60 min each. WC, body fat percentage, fat-free mass, HGS dominant hand, HGS non-dominant hand, 30-s chair stand, and TUG were assessed. All measurements were taken in the afternoon, between 16:00 and 18:00 h, and in the same place (a community center in Osorno, Chile). At the same time, the training sessions (EBT, MCT, and GBD) were carried out between 16:00 and 17:00 h. The older women did not present musculoskeletal and/or cardiorespiratory injuries during the intervention and did not present pain prior to the assessments or during the training sessions.

### 2.2. Participants

Thirty-six physically inactive older women initially participated in the intervention. The sample size calculation indicates that the ideal number of participants per group is 10. According to a previous study [12,16], the lowest difference required for significant clinical relevance was determined to be a mean difference of 0.50 s in the TUG, with a standard deviation of 0.93 s. An alpha level of 0.05, 85% power, and a 20% anticipated loss were taken into account. The GPower program (version 3.1.9.6, Franz Faul, Universiät Kiel, Kiel, Germany) was used to calculate the statistical power. The following were the inclusion criteria: (i) Older women, 60–65 years of age; (ii) those who demonstrate the capacity to comprehend and execute straightforward commands in a contextualized way; (iii) those who are independent, defined as having at least a 43-point score on the Chilean Ministry of Health’s Preventive Medicine Examination for Older People [17]; and (iv) those with the ability to comply with the attendance requirement of at least 85% to the sessions scheduled for the intervention. Regarding the criteria for exclusion, the following were taken into account: (i) being afflicted with a disability of any kind; (ii) being musculoskeletal injured or receiving physical rehabilitation therapy, which keeps them from engaging in their regular physical activity; and (iii) being incapable of engaging in physical activity, either temporarily or permanently. In order to be included in the final analysis, participants who fulfilled the inclusion criteria also needed to complete at least 85% of the training sessions and attend all assessment sessions. The inclusion criteria are summarized in Figure 1.

By approving the use of the data for scientific purposes by the signature of an informed consent form, all participants acknowledged the inclusion criteria for the data’s usage and treatment. The protocol was created in accordance with the Declaration of Helsinki and approved by the scientific ethics committee of Universidad Católica del Maule, Chile (Number: N°29-2022).

### 2.3. Anthropometric Parameters and Sociodemographic Assessments

Bipedal height was measured using a stadiometer (Seca model 220, SECA, Hamburg, Germany; accuracy to 0.1 cm) and body weight was calculated using a mechanical scale (Scale-tronix, Chicago, IL, USA; accuracy to 0.1 kg) while wearing the barest minimum of clothing [18]. The baseline characteristics of the sample are presented in Table 1. For this measurement, the guidelines of the International Society for Advances in Kinanthropometry (ISAK) were followed [19].

### 2.4. Waist Circumference (WC)

A fiberglass tape measure was used to measure the WC, which was attached at the height of the last floating rib, after which the older women were instructed to breathe in and then breathe out, and the circumference was marked [18]. For this measurement, the guidelines of the ISAK were followed [19,20].

### 2.5. Body Composition

Using tetrapolar bioimpedance (InBody 570^®^, Seoul, Republic of Korea) and eight tactile point electrodes, the body fat percentage and fat-free mass were determined following previous recommendations [21].

### 2.6. Handgrip Strength (HGS)

Previous recommendations state that HGS was employed [22]. It was found that the best position for the test was seated position, with the wrist and forearm in neutral, the elbow flexed at a 90-degree angle to the side of the body, the spine aligned, and the shoulder in neutral. To conduct the test, a portable dynamometer (Jamar^®^, PLUS+, Sammons Preston, Patterson Medical, Warrenville, IL, USA) was utilized. In order to provide a firm hold on the device and maintain appropriate closure of the metacarpal phalangeal and interphalangeal joints based on hand size, the dynamometer was positioned in the first position, which encourages contact between the thumb and index finger’s first phalanx. Each person had a 120-s recovery before attempting three times with each hand.

### 2.7. Thirty-Second Chair Stand

The 30-s chair stand test [23] counts the number of repetitions performed while sitting on the chair, with arms resting on the chest, for thirty seconds. It was designed to assess the muscular strength of the lower limbs. After three attempts with a recovery of 120 s between each, the best of the three efforts was attained.

### 2.8. Timed Up-and-Go (TUG)

The TUG test was carried out in accordance with previous recommendations [24]. The person must get out of an arm-supported chair, travel a 3-m hallway, turn around, and return to the chair. They must swiftly record the best of three trials after completing them. Two assessors recorded the time using single-beam photocells (Brower Timing System, Draper, UT, USA); the best three attempts were statistically analyzed.

### 2.9. Intervention

The interventions (EBT, MCT, and GBD) were carried out following the protocols of previous studies [11,12,25]. The programs were designed to consist of an 8-week (16-session) warm-up consisting of joint mobility and low-intensity aerobic activities, a 40-min central component (including EBT, MCT, and GBD), and a 10-min cool-down using static flexibility exercises. Table 2 provides an overview of the intervention dosage.

The EBT program is based on previous studies [11,12] that demonstrated that it is safe and effective for older people. Using the TheraBand^®^ elastic band system (Hygenic Corporation, Akron, OH, USA), participants started with a 10-min warm-up with joint mobility and low-intensity aerobic exercises. The colors of the elastic bands (yellow, red, green, blue, black, silver, and gold), each corresponding to a specific tension range, were used to indicate the training loads. The OMNI Resistance Exercise Scale (OMNI-RES) was used to monitor resistance training intensity, which ranged from moderate to vigorous (5 to 8 points) [26]. Six upper limb muscle strength exercises were done (pull-up, pullback, shoulder abduction, biceps curl, triceps, forearm) and 6 for the lower limbs (leg press, ankle eversion, ankle dorsiflexion, knee extension, knee flexion, and hip flexion). The older women started with the lowest resistance (yellow color), achieving a 10-repetition maximum (10 RM) of an upper and/or lower limbs exercise, moving to the next rubber band color until all 10 RM could not be produced. The training program started with the selection of the last elastic band. The participants engaged in two sets of exercises at 100% (10 RM) intensity throughout each training session, interspersed with one-min rest intervals. Every upper and lower limb exercise was performed twice, with 10 to 15 repetitions each set, for the duration of the 8-week intervention. With an elastic band that offered more resistance, the maximum strength (at 10 RM) was determined. The resistance band length was halved and continued for 8 weeks of intervention if they were unable to advance to a higher resistance band. If they were successful in achieving 10 RM, they continued in color every four weeks.

The MCT program was based on a previous study [25]. The development of the session or central part consisted of a 40-min circuit of distributed work that included cardiorespiratory fitness, agility, and postural balance exercises using elastic bands, poles, 2 kg medicine balls, and chairs. The exercises targeted the biceps, triceps, deltoids, latissimus dorsi, quadriceps, hamstrings, glutes, and gastrocnemius, which correspond to the large muscles of the upper and lower limbs. The first training volume (the first 4 weeks) consisted of three sets of 10 repetitions of each muscle activity, with a 2-min rest between sets. Slow movements were used, with a duration of 2 s for concentric contractions and 4 s for eccentric contractions. Volume was increased to 4 sets of 10 repetitions of each muscle exercise with a 2-min rest between sets (between weeks 5 and 8). The OMNI-RES was used to monitor resistance training intensity, which ranged from moderate to vigorous (5 to 8 points) [26].

The GBD protocol was based on a previous study [12]. It began with a warm-up with dances from the 60s and 70s of low to moderate intensity (10-min). The central part (40 min) consisted of moderate- to high-intensity dances from different epochs (from the 70s to the present), where each song lasted approximately 3 min, with 2 min of recovery between each piece. The intensity remained moderate to vigorous, with participants having a heart rate <120 beats per minute (11). To finish, there was a 10-min cool-down with relaxing music and static flexibility exercises. The assessments and regular intervention sessions (EBT, MCT, and GBD) are summarized in Figure 2.

### 2.10. Statistical Analysis

GraphPad Prism version 9.0 statistical software was used to analyze descriptive and inferential data. The descriptive statistics included the calculation of the mean and standard deviation. The Shapiro-Wilk test was applied to determine the data distribution. Subsequently, a two-factor mixed ANOVA model with repeated measures was performed to measure the time × group effect of the WC, body fat percentage and fat-free mass, the HGS of the dominant hand, the HGS of the non-dominant hand, 30-s chair stand, and TUG. When the time × group interaction was significant, a Bonferroni multiple comparisons test (post hoc) was performed to establish intra-group (pre vs. post assessments) and inter-group (EBT vs. MCT vs. GBD) differences. To determine the effect size of the time × group interaction, the partial eta squared (ηp^2^) was calculated, which was interpreted considering the ηp^2^ values of 0.01, 0.06, and 0.14, which correspond to effect sizes small, moderate, and large, respectively [27]. For multiple comparisons, the effect size was calculated with Cohen’s d [28], considering a small (≥0.2), moderate (≥0.5), or large (≥0.8) effect. For all analyses, an α value of 0.05 was considered.

## 3. Results

Table 3 shows the variables’ before and after intervention results for the EBT, MCT, and GBD. The two-way mixed ANOVA test revealed a significant time × group interaction for body fat percentage (F_(2,27)_ = 4.447; *p* = 0.027; ηp^2^ = 0.330, *large effect*), HGS of the dominant hand (F_(2,27)_ = 11.370; *p* < 0.001; ηp^2^ = 0.558, *large effect*), HGS of the non-dominant hand (F_(2,27)_ = 13.890; *p* < 0.001; ηp^2^ = 0.606, *large effect*), and 30-s chair stand test (F_(2,27)_ = 5.054; *p* = 0.018; ηp^2^ = 0.322, *large effect*). On the other hand, for fat-free mass (F_(2,27)_ = 1.906; *p* = 0.177; ηp^2^ = 0.175, *large effect*), WC (F_(2,27)_ = 1.138; *p* = 0.342; ηp^2^ = 0.112, *medium effect*), and TUG (F_(2,27)_ = 0.604; *p* = 0.557; ηp^2^ = 0.216, *large effect*), there was no significant interaction.

The results of the multiple intragroup and intergroup comparisons are shown in Figure 3. Regarding body fat percentage, significant differences were only found in the MCT group before and after the intervention (*p* = 0.044; ES = 0.426, *small effect*), with significant differences in favor of the EBT group (*p* = 0.001; ES = 2.488, *large effect*). A significant increase in HGS of the dominant hand (*p* < 0.001; ES = 0.977, *large effect*), HGS of the non-dominant hand (*p* < 0.001; ES = 0.583, *moderate effect*), and 30-s chair stand test (*p* = 0.003; ES = 1.612, *large effect*) is observed only in the EBT group. In the HGS non-dominant hand, there was only a significant difference in favor of the EBT group regarding MCT (*p* = 0.044; ES = 0.158), while no significant difference was reported between EBT versus GBD and MCT versus GBD.

## 4. Discussion

This study analyzed the effects of EBT versus MCT and GBD on WC, body composition (body fat percentage and fat-free mass), and physical performance (HGS, 30-s chair stand, and TUG) outcomes in Chilean older women. Among the primary outcomes were verified significant differences in body fat percentage in favor of MCT regarding EBT, and HGS of the non-dominant hand in favor of EBT regarding MCT. However, only the MCT significantly decreased body fat percentage, while only the EBT significantly increased performance in HGS dominant and non-dominant hands and the 30-s chair stand test. The GBD did not report significant changes.

No significant improvements in fat-free mass were reported in EBT, MCT, and GB in contrast to what was reported by Valdés-Badilla, et al. [12] in an intervention using EBT in older women with sarcopenia for 12 weeks with a frequency of 3 sessions per week of 60 min duration, which detected significant increases (*p* < 0.001) in fat-free mass compared to a GBD. However, in a study conducted by Oh, et al. [29] in an 8-week intervention, EBT, with a frequency of 2 sessions per week of 60 min, no significant increases (*p* = 0.60) in fat-free mass were reported compared to an inactive control group in older people. Similarly, Valdés-Badilla, et al. [25] using an MCT intervention for 8 weeks with a frequency of 3 sessions per week of 60 min did not report significant improvements (*p* = 0.62) in fat-free mass compared to an adapted taekwondo group. The contradiction between the studies mentioned above may be due, first of all, to the duration of the interventions; Valdés-Badilla, et al. [12] obtained increases in fat-free mass after 12 weeks of training with a frequency of 3 times per week, which comprises a total of 36 training sessions, compared to Oh, et al. [29] and the present study with 8 weeks of training and a frequency of 2 times per week, comprising 16 training sessions and performing less than half of the sessions carried out [12]. In this sense, the increase in fat-free mass has been shown to follow a dose-response relationship, with increasing gains with higher training volumes [30]. Therefore, if the goal were to maximize muscle growth, a greater amount of weekly time would need to be allocated to achieve this goal. Secondly, it is also essential to take into account the nutritional intake of the participants during the study since the combination of protein intake plus physical exercise is the most efficient strategy to promote the increase in fat-free mass and muscle remodeling [31]. Given that this variable was not controlled in both studies, there is a probability that the food intake in both studies during the interventions may have been different.

Another of the results reported in the body composition variables was the body fat percentage, for which no significant decreases in EBT and GBD were reported but significant decreases in MCT were reported; this is similar to that reported by Lee, et al. [32] in older women with osteosarcopenia, in which no significant decrease (*p* = 0.40) in body fat percentage was reported following a 24-week EBT intervention with a frequency of 3 sessions per week compared to an active control group. However, a study by Huang, et al. [33] in older women with sarcopenia reported significant decreases (*p* < 0.01) in body fat percentage following an intervention using EBT for 12 weeks with a frequency of 3 sessions per week compared to an inactive control group; this is similar to that reported by xMonteiro, et al. [34] in older women, showing significant differences (*p* < 0.05) in body fat percentage in favor of MCT after 32 weeks of training compared to an inactive control group. It is crucial to consider the characteristics of the subjects when analyzing the results since the reduction in the body fat percentage found in Huang, et al. [33] through the use of EBT could be explained by the obesity of older women, who could have been more susceptible to reducing this variable due to their anthropometric characteristics compared to the sample of the present study [35]. On the other hand, MCT, through an exercise program that combines endurance, muscle strength, balance, and agility, has been demonstrated as an effective regimen to reduce the body fat percentage in older people and prevent the functional deterioration associated with age [36]. However, it is essential to mention that the improvements in body composition achieved through an MCT program are not maintained over time. Therefore, although MCT is shown to be an effective exercise-based strategy to improve body composition [36], it should be taken into account that these results tend to decrease in a short period due to the physiological processes of aging [37], such as sarcopenia, increased body fat, and low bone density.

No significant decreases were reported in the WC variable in the EBT, MCT, and GBD, which is similar to that reported by Bong and Song [38] in older women, where no significant decrease (*p* = 0.59) in WC was found with a 12-week EBT intervention with a frequency of 3 sessions per week compared to an inactive control group. However, in a study conducted by Gargallo, et al. [10] using MCT and EBT combined with muscle power exercises for 12 weeks with a frequency of 2 sessions per week of 70 to 80 min per session, a significant decrease (*p* < 0.001) in WC was reported in both interventions compared to an inactive control group; this could be explained by the execution of active rest in Gargallo, et al. [10], where they performed coordinative movements combined with cognitive tasks (e.g., rhythmic limb swinging, psychomotor reaction time exercises, dance movements, task sequences while inverting or “dispersing” an order), which likely contributed to greater energy compared to our study and, therefore, improvements in morphological variables [10].

One of the results found in the present study was a significant increase in HGS non-dominant hand by the EBT compared to MCT, which is similar to that reported by Yang, et al. [39] in community-dwelling older people who presented significant increases (*p* < 0.001) in HGS in dominant and non-dominant hands through an EBT for 32-week with a frequency of 3 sessions per week of 60 min compared to an inactive control group. Similarly, Choi, Hurr, and Kim [8] reported significant (*p* < 0.05) increases in HGS in dominant and non-dominant hands in older people using an EBT for 12 weeks with a frequency of 3 sessions per week of 60 min compared to an inactive control group. These results are similar to those reported by Valdés-Badilla, et al. [12], which show significant increases (*p* < 0.01) in HGS in dominant and non-dominant hands through an EBT compared to a GBD in older women with sarcopenia. The improvements in the EBT group could be explained by the particularities of training with elastic bands, given that they allow different degrees of tension to be applied throughout the entire range of motion [40]. In general, the resistance imposed by the elastic band is low at the beginning of the movement, allowing a resisted execution at high speed, improving the conduction speed of the motor units [41]. However, as the range of movement improves, it increases resistance, forcing subjects to try to maintain muscle power in higher overload ranges [42]. In this sense, the great variety of stimuli provoked by elastic bands would favor brain adaptations that enhance the various expressions of muscle strength [43]. In addition, increases in HGS may result from repeated hand muscle stimulation through exercises requiring repeated acts of gripping and traction [44].

Another result reported on lower body muscle strength was the 30-s chair stand test, where a significant increase in repetitions was reported only in the EBT group. Similar to what was reported by Souza, et al. [45] in older people who underwent EBT for 14 weeks with a frequency of 2 weekly sessions of 60 min compared to traditional resistance training with free weights, significant improvements (*p* < 0.05) were recorded in both groups in the 30-s chair stand test. These are similar results to those reported by Yang, et al. [39] in older people who underwent a 72-week intervention with EBT versus the inactive control group, showing significant improvements (*p* = 0.000) in the 30-s chair stand in favor of the EBT group. The 30-s chair stand test aims to assess dynamic and explosive lower body muscle strength, which is necessary for numerous tasks in daily life [46]. In this sense, concerning our findings, the improvements found in the EBT group can be explained by the performance of lower body exercises that included knee flexion and extension movement at an intensity equivalent to 100 of 10-repetition maximum. This action is the main one during the execution of the 30-s chair stand test, so the muscle strength gains of the lower body can be seen transferred specifically to the performance of the test [47]; in addition, Nakatani, et al. [48] reported a moderate correlation (*r =* 0.52) between the performance of the 30-s chair stand test and the maximum voluntary isometric contraction in knee extension in older women. This was similar to what was reported by Jones, Rikli, and Beam [46], who found a moderately high correlation (*r* = 0.71) between the performance of the 30-s chair stand test and the maximum performance in a weight-adjusted leg press (kg).

On the other hand, no significant differences were reported in the EBT, MCT, and GBD in TUG. In the study by Valdés-Badilla, et al. [25], no significant improvements (*p* = 0.11) were shown in TUG in an MCT compared to an adapted taekwondo intervention in older women. However, in the Miranda-Aguilar, et al. [11] study in older people, a significant reduction time (*p* = 0.04) in TUG was reported in an EBT group compared to a traditional gym equipment group. This is similar to what was reported by Valdés-Badilla, et al. [12] in older women with sarcopenia, showing significant improvements (*p* < 0.001) in the performance of the TUG through EBT compared to a GBD. In a previous study, MCT may have been more effective in increasing the capacity of the extensor muscles of the lower body to generate torque during walking compared to EBT and GBD [49], which translates into improvements in the reactive capacity of the lower body generating positive changes in TUG [50].

### Strengths and Limitations

Potential limitations of this study include: (i) lack of control over food intake and failure to complete a food record (to determine participants’ eating habits, including their macronutrient and micronutrient intake), factors that could affect the morphological variables and physical performance of the participants; however, participants were instructed to maintain their eating habits during the intervention; (ii) the lack of inclusion of male participants does not provide the possibility of generalizing these results for the entire population and comparing them between the male and female sex; (iii) considering only apparently healthy older women who completed ≥ 85% of all training sessions, which could limit the analysis; (iv) not analyzing cardiovascular and/or metabolic variables; (v) not performing a follow-up analysis; (vi) not conducting an intent to treat analysis, and; (vii) the results in body composition may have been affected by baseline differences in the groups analyzed. The strengths include: (i) the randomized controlled design with three groups that allows the effectiveness of the interventions to be better determined; (ii) an active control group such as GBD; (iii) the training dose was similar in the three groups, and; (iv) the inclusion of a specific exercise for the wrist flexor-extensor muscles in EBT, which led to statistically significant improvements in HGS. Future studies could analyze the effect of these interventions on physiological and psychoemotional variables in older men and women and try to understand the qualitative approach of the preferences and wishes of older people to physical activity practice. Finally, considering the positive responses of the present study, research with larger groups in different places, cities, and countries could be conducted to understand possible positive responses to public health policies.

## 5. Conclusions

MCT significantly reduced the body fat percentage regarding EBT, and EBT significantly improved the HGS of the non-dominant hand regarding MCT, with no differences reported in the rest of the analyzed variables between the groups. Therefore, selecting appropriate and well-dosed exercises can improve HGS in older women through an effective, economical, and easily accessible method such as elastic bands.

## Figures and Tables

**Figure 1 life-14-01362-f001:**
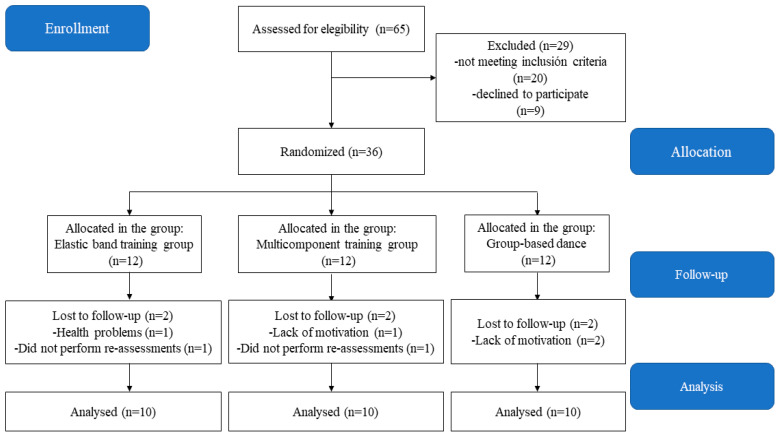
Study flowchart of the enrolment process, allocation, follow-up, and analysis of older women.

**Figure 2 life-14-01362-f002:**
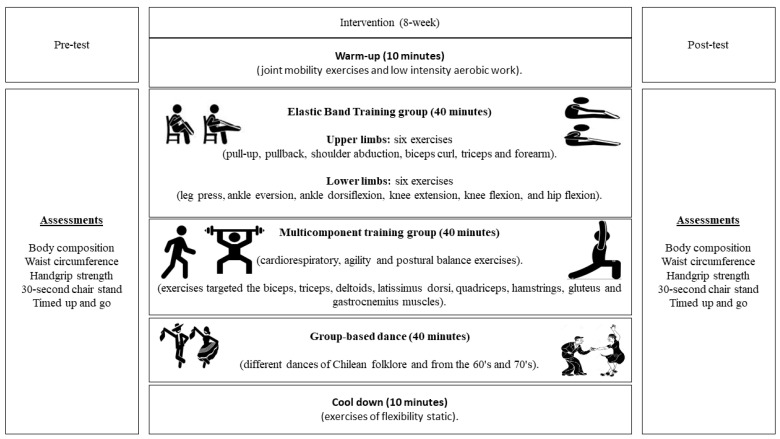
Assessments and regular sessions of the intervention.

**Figure 3 life-14-01362-f003:**
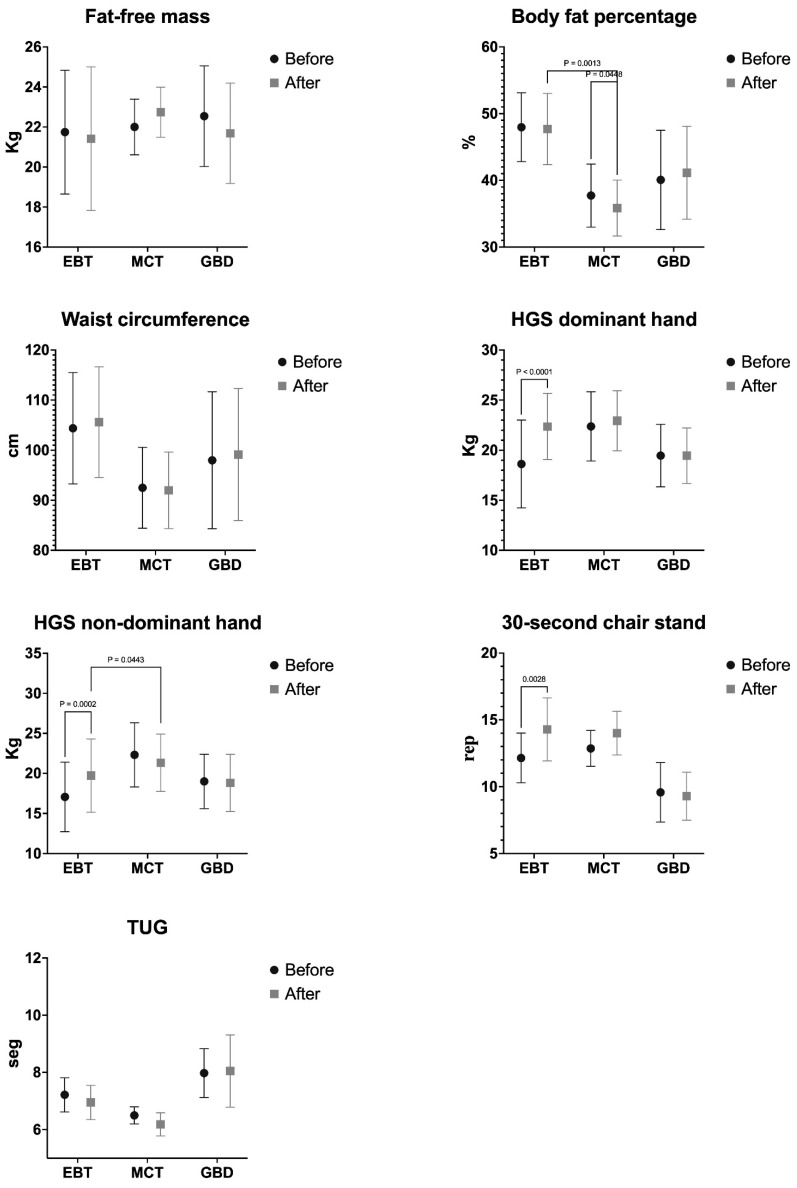
Multiple intragroup and intergroup comparisons of elastic band training versus multicomponent training and group-based dance on body composition and physical performance. Note: HGS: handgrip strength; TUG: timed up-and-go; EBT: elastic band training; MCT: multicomponent training; GBD: group-based dance.

**Table 1 life-14-01362-t001:** Baseline anthropometric parameters and sociodemographic assessments of older women.

Variable.	Assessment	EBT Group (*n* = 10)	MCT Group (*n* = 10)	GBD (*n* = 10)
Age (years)		70.2 (5.76)	72.0 (4.39)	74.4 (4.03)
Anthropometric parameters	Bipedal height (cm)	1.48 (0.05)	1.53 (0.05)	1.51 (0.05)
	Body mass (kg)	71.4 (39.4)	71.1 (14.5)	65.5 (4.99)
Academic level	Primary (%)	15	13	15
	Secondary (%)	13	15	12
	Bachelor (%)	4	8	5
	Postgraduate (%)	0	0	0
Civil status	Married (%)	28	10	10
	Separated (%)	8	10	6
	Widowed (%)	2	4	2
	Single (%)	0	0	0

Note: EBT: elastic band training; MCT: multicomponent training; GBD: group-based dance; Data are presented in mean and standard deviation.

**Table 2 life-14-01362-t002:** Intervention dosage.

Program	Duration (Weeks)	Frequency (Weekly)	Time per Session (min)	Physical Exercise	Sets	Repetitions	Recovery	Intensity
EBT	1–4	2	60	Upper limb	2	10–15	2 min	OMNI-RES (5–8 points)
	5–8			Lower limb				
MCT	1–4		60	Upper limb	3	10	2 min	OMNI-RES (5–8 points)
	5–8			Lower limb	4			
GBD	1–4		60	Whole body	8	3 min	2 min	Heart rate < 120 beats
	5–8							

Note: EBT: elastic band training; MCT: multicomponent training; GBD: group-based dance. OMNI-RES: OMNI Resistance Exercise Scale.

**Table 3 life-14-01362-t003:** Time × group interaction in the analyzed variables of elastic band training versus multicomponent training and group-based dance on body composition and physical performance.

	Group	Before		After		Time × Group *p*-Value	Time × Group F-Value	ηp^2^	Classification
		Mean	SD	Mean	SD				
Fat-free mass (kg)	EBT	21.7	3.1	21.4	3.6	0.177	1.906	0.175	*Large effect*
	MCT	22.0	1.4	22.7	1.2				
	GBD	22.5	2.5	21.7	2.5				
Body fat percentage (%)	EBT	48.0	5.2	47.7	5.3	**0.027 ***	4.447	0.330	*Large effect*
	MCT	37.7	4.7	35.8	4.2				
	GBD	40.1	7.4	41.1	7.0				
Waist circumference (cm)	EBT	104.4	11.1	105.6	11.0	0.342	1.138	0.112	*Medium effect*
	MCT	92.5	8.1	92.0	7.6				
	GBD	98.0	13.7	99.1	13.2				
HGS dominant hand (kg)	EBT	18.6	4.4	22.4	3.3	**<0.001 *****	11.370	0.558	*Large effect*
	MCT	22.4	3.5	22.9	3.0				
	GBD	19.5	3.1	19.5	2.8				
HGS non-dominant hand (kg)	EBT	17.1	4.3	19.7	4.6	**<0.001 *****	13.890	0.606	*Large effect*
	MCT	22.3	4.0	21.3	3.6				
	GBD	19.0	3.4	18.8	3.6				
30-s chair stand (rep)	EBT	12.1	1.9	13.3	1.9	**0.018 ***	5.054	0.322	*Large effect*
	MCT	12.9	1.3	11.7	1.0				
	GBD	9.6	2.2	9.3	1.8				
TUG (seg)	EBT	7.6	0.6	7.2	0.5	0.557	0.604	0.216	*Large effect*
	MCT	7.8	0.3	6.5	1.1				
	GBD	8.0	0.9	8.1	1.3				

Note: SD: standard deviation. HGS: handgrip strength; TUG: timed up-and-go; ηp^2^: partial eta square. EBT: elastic band training. MCT: multicomponent training. GBD: group-based dance; * = *p* < 0.05; *** = *p* < 0.001.

## Data Availability

The datasets generated during and/or analyzed during the current research are available from the corresponding author upon reasonable request.

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
