# Peer review of "Elastic Band Training Versus Multicomponent Training and Group-Based Dance on Morphological Variables and Physical Performance in Older Women: A Randomized Controlled Trial"

_life, 2024, doi:10.3390/life14111362_

Round 1

Reviewer 1 Report

Comments and Suggestions for Authors

Dear Authors,

Interesting study addressing the differences on physical outcomes for older women based on three different types of training. Elastic band training, multicomponent training and group-based dance are common among older people and are appropriate training strategies.

The introduction and methods are very well written and figures and descriptions are very well done. Results and discussion are appropriate for the methods. The only concern I have is the conclusion as is restates the discussion and is very hard to read as the conclusion should be a simple summary of the findings. Reading it as if this and that makes it hard to read and had to referee back to the graphs to make sure I read it right. State was EBT was better overall as it showed the greatest improvement in these areas, however MCT produced positive results in these areas while we do not recommend GBD as it showed little to no improvement in any area of performance that we measured.

Overall good work and a study for a very needed population.

Reviewer 2 Report

Comments and Suggestions for Authors

Working with older adult population groups is increasingly important to improve the quality of life of this population as well as to delay the onset of certain pathologies associated with aging and continue to maintain good levels of autonomy.

That is why the authors propose a study of great interest for this population group and how activities with high motivation for the elderly can be beneficial in variables such as those presented by the authors, of morphological and physical character.

In reference to the introductory section, the authors present the development of this section in a clear manner, coherent with the objective they establish and using a very good percentage of references from the last 5 years. This guarantees the suitability of the content presented in this section. In addition, the objective and hypothesis of the research are stated, although it would have been desirable that other research questions had been posed.

In reference to the methods section, the authors clearly and thoroughly detail the method used in their research. In addition, they present figures (Figure 1) that facilitate understanding of the process.

Likewise, they clearly detail the inclusion and exclusion criteria of the sample participating in this study. In relation to the sample, a strength of the research is the detail that the authors present on the sample from subsection 2.3, being detailed, clear and extensive and including tables and figures that facilitate the interpretation of the data presented at the anthropometric, test and intervention levels.

In the same line, the results presented by the authors are consistent with the proposed statistics and again the tables and figures presented facilitate their interpretation.

Finally, the discussion and in the same line as the introduction of the work, presents a wide range of current citations that support this section, address the limitations of the study as well as its strengths, which can be a starting point for future research in this line of work.

For all these reasons and for the quality presented in the work, we recommend the publication of this work once minor corrections have been made, especially in the use of English and formal aspects.

Comments on the Quality of English Language

Minor editing of English language required.

Reviewer 3 Report

Comments and Suggestions for Authors

First of all, the reviewer would like to thank the authors for their work and efforts in trying to improve  sport science  knowledge.

The article is of interest but needs some changes to be made before it can be considered for publication.

Abstract

The results should not only indicate between which groups the results were obtained, but also indicate in which direction. For example, “the EBT group scored better on the HGS than the MCT group” or whatever applies.

_Keywords:

Highlighting should be eliminated from both keywords and the rest of the text.

Introducción:

Line 50: authors should review the citation rules of the journal. They should do this with the rest of the text.

Methods:

Line 105: there is an error in the paragraph break.

Line 116: Figure 1 does not summarize the inclusion criteria. This should not be put in the text. In addition, the figure should detail what inclusion criteria were not met for each subject to not be included.

Did the authors take into consideration the previous sporting experience of the participants, how did they monitor physical activity outside the program, and did they instruct the participants not to engage in any other practice? If not, these data should be added in the analysis and included in the study limitations.

Line 167: what measures were taken following ISAK standards. This society provides training on anthropometric methods such as weight, height, skinfolds, diameters. And here it is indicated that body composition was calculated with bioimpedance.

170: the translation should be revised. “sedentary” is not a position. Perhaps they mean “seated position”.

Results

The authors indicate that there are significant differences in fat loss between the MCT and EBT groups, however, Figure 3 clearly indicates that the EBT group had much higher fat percentages at the start of the program than the MCT group. How did they take this into consideration?

References

The list of references should be revised. Errors were detected. I point out some of them

Reference 16 and 19, are incomplete

Round 2

Reviewer 3 Report

Comments and Suggestions for Authors

The manuscript has reached sufficient quality for publication. Thanks to the authors for their efforts